# Lignin-Derived Oligomers as Promising mTOR Inhibitors: Insights from Dynamics Simulations

**DOI:** 10.3390/ijms26178728

**Published:** 2025-09-07

**Authors:** Sofia Gabellone, Giovanni Carotenuto, Manuel Arcieri, Paolo Bottoni, Giulia Sbanchi, Tiziana Castrignanò, Davide Piccinino, Chiara Liverani, Raffaele Saladino

**Affiliations:** 1IRCCS Istituto Romagnolo per lo Studio dei Tumori “Dino Amadori”—IRST Srl, 47014 Meldola, Italy; sofia.gabellone@irst.emr.it (S.G.); giulia.sbanchi@irst.emr.it (G.S.); chiara.liverani@irst.emr.it (C.L.); 2Department of Ecological and Biological Sciences, Tuscia University, Viale dell’Università s.n.c., 01100 Viterbo, Italy; giovanni.carotenuto@studenti.unitus.it (G.C.); d.piccinino@unitus.it (D.P.); saladino@unitus.it (R.S.); 3Department of Computer Science, “Sapienza” University of Rome, V. le Regina Elena 295, 00161 Rome, Italy; manuel.arcieri@uniroma1.it (M.A.); bottoni@di.uniroma1.it (P.B.)

**Keywords:** mTOR, mTOR inhibitors, lignin-derived oligomers, molecular dynamics (MD) simulations, rapamycin, everolimus, MM/PBSA binding free energy, hydrophobic and hydrogen bond interactions, high-performance computing (HPC)

## Abstract

The mammalian target of rapamycin pathway, mTOR, is a crucial signaling pathway that regulates cell growth, proliferation, metabolism, and survival. Due to its dysregulation it is involved in several ailments such as cancer or age-related diseases. The discovery of mTOR and the understanding of its biological functions were greatly facilitated by the use of rapamycin, an antibiotic of natural origin, which allosterically inhibits mTORC1, effectively blocking its function. In this entirely computational study, we investigated mTOR’s interaction with seven ligands: two clinically established inhibitors (everolimus and rapamycin) and five lignin-derived oligomers, a renewable natural polyphenol recently used for the drug delivery of everolimus. The seven complexes were analyzed through all-atom molecular dynamics simulations in explicit solvent using a high-performance computing platform. Trajectory analyses revealed stable interactions between mTOR and all ligands, with lignin-derived compounds showing comparable or enhanced binding stability relative to reference drugs. To evaluate the stability of the molecular complex and the behavior of the ligand over time, we analyzed key parameters including root mean square deviation, root mean square fluctuation, number of hydrogen bonds, binding free energy, and conformational dynamics assessed through principal component analysis. Our results suggest that lignin fragments are a promising, sustainable scaffold for developing novel mTOR inhibitors.

## 1. Introduction

PI3K/AKT/mTOR (PAM) is an important signaling network in eukaryotic cells associated with survival, growth, and proliferation [1]. This pathway is frequently dysregulated in human cancers, with aberrations observed in nearly 50% of cancer patients [2]. mTOR (the mechanistic target of rapamycin) is a central serine/threonine kinase in this network. It integrates nutrient and growth-factor signals to regulate cell growth and metabolism, and it is often the point at which the pathway becomes dysregulated [3,4]. Structurally, mTOR is organized into an N-terminal solenoidal region composed of HEAT (a type of protein structural motif composed of two antiparallel α-helices that stack together in tandem arrays, forming a solenoidal, superhelical scaffold) and FAT repeats (FRAP–ATM–TRRAP domain, a large α-helical region found in all members of the PIKK, phosphatidylinositol 3-kinase–related kinase, family), followed by the FRB domain (FKBP12–rapamycin binding domain), the kinase domain, and a short FATC motif (FAT C-terminal domain) [5]. Starting from the N-terminus, the large FAT α-helical stack (residues 1–629) provides an extended scaffold that stabilizes and shapes the catalytic core; immediately downstream lies the FRB domain (residues 630–763), then the kinase domain (residues 764–1045; PIKK family), and finally the FATC motif (residues 1046–1164), which caps the enzyme and contributes to structural integrity and catalytic competence. As shown in Figure 1, these elements form a continuous structural pathway that transmits regulatory inputs from the FAT/FRB regions to the active site. Functionally, this architecture underpins substrate specificity and allosteric regulation, including inhibition by rapamycin via FKBP12 binding (FK506-binding protein 12) to the FRB domain [6]. mTOR operates within two multiprotein assemblies: mTORC1 (mechanistic target of rapamycin complex 1), which is rapamycin-sensitive and promotes protein synthesis and autophagy, enhancing nutrient uptake [7,8], and mTORC2 (mechanistic target of rapamycin complex 2), which contributes to membrane and mitochondrial integrity [9].

Elevated expression of the mTOR signaling pathway is commonly recognized as a biological marker for breast cancer [10]. This phenomenon is often associated with enhanced activity of ErbB family receptors or with genetic alterations and mutations in the PI3K signaling pathway [11]. mTOR inhibitors, including rapamycin and the semi-synthetic derivative everolimus (EVE), are widely studied and applied in clinical trials for the treatment of cancer, showing specific beneficial effects in hormone receptor-positive (HR+) breast cancers [12]. Despite its clinical relevance, the therapeutic utility of EVE is hindered by its oxidative degradation which may promote resistance and underscore the need for alternative treatment strategies. To overcome these limitations, the development of novel drug delivery systems has gained considerable attention in oncology. In this context, biocompatible and biodegradable lignin-based nanoparticles (LNPs) have emerged as promising candidates for controlled drug delivery, offering stability and enhanced therapeutic efficacy [13,14,15]. Lignin is a complex biopolymer that can be degraded into various oligomeric products characterized by intrinsic therapeutic effects and synergistically enhanced drug activity [16].

Building on previous research [16,17,18], we hypothesize that specific micro-environmental conditions operative in the cancer cells (e.g., acidic conditions and production of hydrogen peroxide) lead to the partial degradation of LNPs, generating oligomers with potential synergistic activity, especially in modulating mTOR pathways. This dual-action mechanism, combining controlled EVE release and the intrinsic mTOR modulatory capabilities of lignin-derived oligomers, may represent a promising “on/off” switch strategy to significantly improve EVE-based cancer therapies, potentially overcoming existing resistance mechanisms and enhancing overall treatment outcomes [18].

To gain insight into the interaction between lignin oligomers and mTOR, we present a detailed explicit-solvent molecular dynamics (MD) simulation of five selected lignin oligomers (moll10-mol14; Table 1). These specific oligomers are well-known molecular fragments generated from lignin during oxidative stress [19,20,21,22,23,24,25]. The results were compared with rapamycin and EVE as references. Molecular dynamics (MD) simulations have become increasingly important in computational chemistry, providing a powerful tool to investigate large and complex macromolecular systems [26,27,28,29,30]. In particular, ligand–receptor interactions have been extensively investigated by MD simulations to capture both binding affinity and time-dependent behavior [31,32,33,34,35,36]. The optimization of MD algorithms for high-performance computing (HPC) platforms further enabled the simulation of biologically relevant timescales, improving the reliability of predictive modeling [37,38]. Furthermore, from the resulting MD trajectories, we carried out Molecular Mechanics Poisson–Boltzmann Surface Area (MM/PBSA) binding free energy calculations, principal component analysis (PCA), and time-resolved ligand–receptor interaction profiling to evaluate the interaction of seven ligand oligomers complexed with mTOR. This study is entirely computational; no experimental binding or cellular assays were performed. The predictions are hypothesis-generating and will guide future biochemical validation, leveraging established computational methods for assessing protein–ligand interactions [39].

## 2. Results and Discussion

In this section, we present and discuss the results of molecular dynamics simulations of the seven ligand–mTOR complexes, carried out in an HPC environment, including (i) ligand and protein-backbone RMSD; (ii) per-residue RMSF with 3D flexibility mapping; (iii) principal component analysis (PCA) of ligand heavy-atom coordinates (PC1–PC2); (iv) time-resolved interaction profiles (hydrogen bonds and hydrophobic contacts) with occupancy metrics; (v) MM/PBSA binding free energy estimates computed over equilibrated windows; (vi) time evolution of ligand–receptor binding; (vii) ADME profiling (SwissADME); and (viii) multi-observable ligand prioritization. This study does not include an apo mTOR MD baseline; all molecular dynamics assessments are confined to ligand-bound complexes.

### 2.1. Classical Molecular Dynamics Simulation

To evaluate the dynamic behavior and structural stability of the seven ligand–mTOR complexes, classical molecular dynamics simulations were carried out over 1000 ns in explicit solvent. The trajectories were analyzed in terms of root mean square deviation (RMSD) and root mean square fluctuation (RMSF), focusing on both the global stability of the complexes and the residue-level flexibility of the protein.

After an initial adjustment (at around 70 ns), the mTOR-backbone RMSD exhibits a slow upward drift, being approximately 0.50 nm over 400–600 ns and reaching around 0.55 nm by 800–1000 ns, whereas the everolimus ligand remains stable around 0.43 nm for the rest of the 1 µs simulation (Figure 2). No abrupt deviations or large conformational jumps are observed.

This behavior suggests that the complex reaches a conformational equilibrium early in the simulation and remains stable. The persistence of the ligand within the binding pocket, without significant displacement, supports a favorable binding mode and highlights the structural compatibility between everolimus and mTOR.

The mTOR–rapamycin complex displays a higher degree of structural deviation compared to the everolimus complex. The RMSD of the protein backbone rises rapidly during the initial 150 ns and stabilizes around 0.70–0.80 nm for the remainder of the simulation (Figure 3). Despite this relatively high average value, the trajectory remains consistent over time without major disruptions. Throughout the simulation, rapamycin demonstrates a remarkably stable RMSD, briefly rising to 0.25 nm before settling at 350 ns to an average of 0.40 nm. This transient fluctuation does not compromise its overall binding stability.

These results suggest that, while the mTOR structure undergoes more pronounced conformational rearrangements in the presence of rapamycin compared to everolimus, the ligand itself remains consistently positioned within the binding site. This behavior confirms its well-established inhibitory role, although the relatively elevated protein RMSD may reflect an adaptive response of the receptor to the ligand’s structural features.

In the mTOR–mol10 complex, the protein exhibits a gradual increase in RMSD during the first 400 ns, stabilizing around 0.52 nm thereafter. Notably, the mol10 ligand maintains a remarkably low and consistent RMSD throughout the entire 1000 ns trajectory, with average values around 0.20–0.25 nm and no significant deviations observed (Figure 4).

This result indicates that mol10 remains tightly anchored within the binding site, exhibiting minimal structural displacement during the simulation. The combination of a stable ligand RMSD and a moderately adjusting protein structure suggests a well-accommodated interaction, possibly reflecting a favorable and rigid binding mode for this lignin-derived compound.

The RMSD profile of the mTOR–mol11 complex reveals a stable behavior of the protein backbone, with values centering around 0.48 nm after the initial equilibration phase of 200 ns. In contrast, the ligand mol11 displays persistent structural variability throughout the simulation, with RMSD values frequently exceeding 0.60 nm and lacking a clear stabilization trend (Figure 5). This pattern suggests that while the protein structure remains relatively steady, mol11 does not adopt a fixed conformation within the binding site. The continuous positional rearrangements of the ligand may indicate a less stable interaction with mTOR, possibly due to suboptimal steric or electrostatic complementarity. Consistent with the large-amplitude, non-plateauing RMSD oscillations observed in Figure 4, ligand mol11 exhibits clear binding instability, repeatedly departing from its initial pose throughout the 1 µs trajectory.

The mTOR–mol12 complex demonstrates a stable RMSD profile for both components. The protein backbone stabilizes around 0.52 nm after the initial 130 ns and after some minor conformational changes it reaches a plateau at 0.60 nm. The ligand mol12 maintains a consistent RMSD near 0.30 nm; at 600 ns the ligand goes through a major conformational change, rapidly reaching a new equilibrium at 0.55 nm (Figure 6).

Overall, the receptor remains remarkably static over the 1000 ns simulation, exhibiting only minimal structural displacement, while the ligand samples two distinct equilibrium conformations. Notably, among the lignin-derived compounds, mol12, despite undergoing a pronounced conformational change, establishes an exceptionally stable complex with mTOR, underscoring its favorable binding properties.

The analysis indicates that the mTOR–mol13 complex exhibits a moderate RMSD profile throughout the simulation. Initially, the protein undergoes a rapid conformational adjustment, after which it gradually stabilizes around 0.55 nm by 300 ns. Concurrently, the ligand, mol13, rapidly achieves stability at 0.35 nm, reaching a local minimum. However, the ligand’s conformation is not static. It experiences a significant conformational change at 700 ns, increasing its deviation to 0.60 nm, followed by a decrease to 0.4 nm at 800 ns. In the final 50 ns of the simulation, mol13 undergoes another conformational shift, raising its deviation to 0.50 nm (Figure 7).

The observed RMSD profile suggests that mol13 initially achieves a conformationally stable state within the binding pocket. However, this stability appears transient, as the ligand subsequently exhibits more dynamic behavior. Compared to other lignin derivatives, mol13’s rapid initial stabilization, followed by a prolonged search for a new equilibrium, may indicate a more flexible or perhaps less optimal initial binding pose.

The mTOR–mol14 complex demonstrated distinct stability profiles for its components. The protein subunit exhibited a robust stability, rapidly converging to an RMSD of approximately 0.45 nm within the first 50 ns of the simulation and maintaining this conformation thereafter. In contrast, the ligand mol14 displayed a trend of increasing stability over the simulation’s duration. It underwent several conformational changes in the initial stages, ultimately reaching an optimal RMSD of 0.48 nm by 400 ns, after which it maintained this stable plateau for the remainder of the simulation (Figure 8).

This observed gradual increase in the ligand’s RMSD suggests a progressive rearrangement of mol14 within the binding pocket. This could indicate either a partial repositioning or an enhanced conformational stability of the ligand relative to the protein. Although the overall mTOR–mol14 complex maintained stability, mol14 displayed a more dynamic binding mode. This dynamic behavior ultimately contributed to a more stable final conformation for the ligand compared to other lignin-derived compounds.

When comparing the RMSD behavior across all seven mTOR–ligand complexes, notable differences in ligand stability and binding dynamics emerge. Among the known inhibitors, everolimus exhibited the most stable profile for both protein and ligand, while rapamycin showed a relatively higher protein RMSD, possibly reflecting broader conformational adaptation by mTOR.

Among the lignin-derived ligands, mol10 and mol12 stood out for their highly stable ligand RMSD values, with mol10 remaining stable for the whole simulation and mol12 undergoing just one major conformational change, suggesting strong and well-defined binding within the pocket. On the contrary, mol11 displayed the least stable ligand behavior, with continuous positional changes throughout the simulation, indicating a weaker or less specific interaction. The ligand mol13 displayed intermediate behavior, characterized by dynamic conformational changes occurring in the latter portion of the simulation. The ligand mol14 exhibited a particularly noteworthy behavior, achieving stability in the latter half of the simulation. Crucially, this occurred while the receptor maintained an exceptionally stable RMSD profile throughout the entire simulation.

Overall, these RMSD results highlight mol10, mol12, and mol14 as the most promising lignin-based ligands in terms of conformational stability during molecular dynamics, potentially correlating with higher binding stability or persistence within the active site.

mTOR is a serine/threonine kinase. Focusing on chain B of the mTOR protein, we can identify several key regions contributing to its overall architecture and function. The N-terminal region, spanning approximately residues 1-591, comprises the FAT (FRAP–ATM–TRRAP) domain. This domain is characterized by stacked alpha-helices, forming a rigid central core that is thought to mediate protein–protein interactions and provide structural scaffolding. Following this is the FRB domain, located between residues 630 and 729, which serves as the direct target of the rapamycin–FKBP12 complex, a critical regulatory interaction. Further downstream lies the highly conserved kinase domain, spanning residues 764–1045. As a member of the PIKK (phosphoinositide 3-kinase-related kinase) family, this catalytic domain is responsible for mTOR’s essential phosphorylation activity. Finally, the C-terminus features the FATC domain (residues 1046–1164), a relatively short yet important region that contributes to structural stability and influences the kinase’s catalytic function. Together, these domains form the functional architecture of mTOR chain B, enabling its complex regulatory roles in cellular signaling.

To evaluate the local flexibility of mTOR in the presence of different ligands, we conducted a root mean square fluctuation (RMSF) analysis on the backbone atoms of chain B (residues 1–1164). A threshold of 0.5 nm was adopted to identify regions with enhanced dynamic behavior.

In the RMSF profiles for the rapamycin and everolimus complexes (Figure 9), seven fluctuation peaks (R1–R7) were detected, primarily located in the FAT domain (residues 1–591), FRB domain (residues 630–729), and FATC domain (residues 1046–1164). Among these, region R6 (corresponding to residues 434–471 and 478–480) exhibited the most pronounced flexibility. Region R7, located at the edge of the FRB domain (residues 706–709), also surpassed the fluctuation threshold.

mTOR dysregulation is involved in several ailments such as cancer or age-related disease. The discovery of mTOR and the understanding of its biological functions were greatly facilitated by the use of rapamycin, an antibiotic of natural origin, which allosterically inhibits mTORC1, effectively blocking its function. In this study, we investigated mTOR’s interaction with seven ligands: two clinically established inhibitors (everolimus and rapamycin) and five lignin-derived oligomers, a renewable natural polyphenol recently used for the drug delivery of everolimus. The seven complexes were analyzed through all-atom molecular dynamics (MD) simulations in explicit solvent using a high-performance computing (HPC) platform. Trajectory analyses revealed stable interactions between mTOR and all ligands, with lignin-derived compounds showing comparable or enhanced binding stability relative to reference drugs. To evaluate the stability of the molecular complex and the behavior of the ligand over time, we analyzed key parameters including root mean square deviation (RMSD), root mean square fluctuation (RMSF), number of hydrogen bonds, binding free energy calculated via the MM/PBSA method, and conformational dynamics assessed through principal component analysis (PCA). Our results suggest that lignin fragments are a promising, sustainable scaffold for developing novel mTOR inhibitors.

For the complexes with the five novel ligands (mol10 to mol14), eight fluctuation regions (R1–R8) were observed (Figure 10). In these systems, the same seven regions described above were consistently observed, with an additional region R8 (residues 1068–1097) in the FATC domain showing significant fluctuations in several complexes.

Importantly, the central kinase domain (residues 764–1045) remained relatively rigid in all simulations, underscoring its structural stability. Conversely, the N- and C-terminal segments, especially within the FAT and FATC domains, demonstrated greater flexibility, likely associated with their roles in regulatory interactions and conformational adaptation.

The structural superposition of RMSF data onto the 3D model of mTOR highlights these dynamic regions (in red), confirming that the most mobile residues are located in peripheral helices and loops. These flexible segments may represent functional hotspots modulated by ligand binding and could be critical for allosteric regulation of the mTOR complex.

Moreover, the region R6, which exhibits the greatest flexibility, was not present in the crystallographic PDB input file. Its reconstruction via predictive homology modeling is therefore expected to result in greater mobility, as flexible regions are typically challenging to capture accurately using crystallographic techniques.

### 2.2. MM/PBSA Binding Free Energy Analysis

The binding free energies (ΔG) of the seven ligand–mTOR complexes were estimated using the MM/PBSA method, based on 8000 frames extracted from the final 800 ns of each molecular dynamics trajectory. Among the lignin-derived ligands, mol14 exhibited the most favorable binding free energy (−37.76 ± 3.41 kcal/mol), followed by mol13 (−31.02 ± 7.51 kcal/mol), mol10 (−26.37 ± 4.84 kcal/mol), mol11 (−25.93 ± 6.44 kcal/mol), and mol12 (−25.12 ± 7.54 kcal/mol). The reference inhibitors showed ΔG values of −26.39 ± 6.14 kcal/mol for rapamycin and −31.51 ± 9.61 kcal/mol for everolimus.

These results highlight mol14 as the ligand with the strongest predicted binding affinity to mTOR, surpassing even the clinically established inhibitors. Interestingly, mol13 and mol14 displayed ΔG values comparable to or more favorable than everolimus, suggesting that these lignin-derived compounds may achieve similar or improved binding performance. On the other hand, mol11 and mol12 yielded the weakest binding energies among the tested ligands, which is consistent with their higher structural variability observed during RMSD analysis.

Overall, the MM/PBSA calculations reinforce the potential of specific lignin oligomers, particularly mol14, as promising candidates for mTOR inhibition, warranting further investigation and validation.

### 2.3. Principal Component Analysis

To assess the collective motions and dominant conformational changes of the protein–ligand complexes, principal component analysis (PCA) was performed on the heavy atoms (carbon and oxygen atoms) of the ligand trajectories for each system throughout the 1000 ns simulation, a standard technique for identifying essential dynamics from simulation trajectories [40]. The projection of the trajectories onto the first two principal components (PC1 and PC2) allows a visual representation of the conformational space sampled by each complex (see Figure 11). A global comparison across all systems is also provided in the combined plot (Figure 11h), which highlights the relative distribution and compactness of each ligand–receptor trajectory.

The mTOR–everolimus complex (Figure 11a) displays a compact clustering of frames with limited dispersion showing mainly three well-defined clusters, suggesting exploration of a narrow conformational space and a stable dynamic profile of the ligand throughout the trajectory. Similarly, the mTOR–rapamycin system (Figure 11b) reveals a dense core cluster with only mild spreading with two main conformations, consistent with an overall stable conformation and limited large-scale motion.

Among the lignin-derived ligands, the mTOR–mol10 complex (Figure 11c) shows narrower distribution along the principal component axes compared to the reference compounds, indicating decreased conformational sampling while remaining within a defined region. In contrast, the mTOR–mol11 complex (Figure 11d) displays the widest dispersion across PC1 and PC2, with multiple scattered regions suggesting a high degree of conformational variability and structural flexibility. The mTOR–mol12 complex (Figure 11e) exhibits an intermediate behavior, forming two moderately compact clusters with peripheral dispersion, indicative of a dynamic equilibrium between related conformational states.

In comparison, the mTOR–mol13 complex (Figure 11f) forms a widespread and fragmented cluster, showing dynamic movement along the principal components, which reflects an unstable structural ensemble. Notably, the mTOR–mol14 complex (Figure 11g) presents the most compact PCA projection among all ligands tested, with tightly grouped frames and restricted conformational drift. The distinct ring-like distribution of points indicates that mol14 actively explores a range of conformations, suggesting significant internal flexibility rather than a single rigid binding pose. This highlights that the ligand’s RMSD plateau observed in later simulation stages represents a stable average position around which considerable dynamic rearrangements occur, reinforcing its role as the most stabilizing ligand in this study. In this study, PC spaces are defined independently for each complex and applied to ligand heavy atoms, so any overlap observed in Figure 11h should be interpreted as a ligand-centric similarity in conformational breadth within the respective pockets rather than evidence of shared protein conformational landscapes.

Overall, the PCA results confirm that rapamycin and everolimus maintain stable dynamic profiles, consistent with their known binding behavior. Among the lignin-based ligands, mol14 and mol10 emerge as the most stabilizing candidates, exhibiting limited conformational sampling and strong clustering, in agreement with their favorable RMSD and binding free energy profiles. On the other hand, mol11 appears as the most dynamically variable system, potentially undermining its effectiveness as a stable mTOR binder. This pattern confirms the instability observed in Figure 5, as evidenced by the persistent, large-amplitude RMSD fluctuations. The remaining lignin ligands, mol12 and mol13, occupy an intermediate position, with PCA results reflecting moderate flexibility in the protein–ligand complexes.

### 2.4. Dynamics of Ligand–Receptor Binding over Time

The molecular modeling analysis has elucidated distinct interaction profiles for rapamycin, everolimus, and several novel ligands, including mol10, mol11, mol12, mol13, and mol14, with the mammalian target of rapamycin (mTOR). In line with these time-resolved contact profiles, the set of major persistent interactions for each complex (threshold > 15% occupancy) is reported in Appendix A, where occupancies are given in parentheses, e.g., Leu801 (35.8%), to facilitate cross-ligand comparison of binding determinants over time. Rapamycin, the benchmark inhibitor, establishes significant hydrophobic interactions with Ile779 (17.7%) and Ile972 (14.0%), as well as Leu801 (8.1%) and Thr861 (6.9%). Concurrently, it forms strong hydrogen bonds with Thr780 (27.1%) and Gln777 (28.0%), consistent with its established binding to the FKBP12–rapamycin binding (FRB) domain. Everolimus, a derivative of rapamycin, displays a different set of hydrophobic contacts, notably with Leu877 (15.4%) and Tyr1158 (21.8%), and forms a key hydrogen bond with Lys986 (26.1%).

The novel ligands exhibit diverse interaction strategies. mol10 engages in robust hydrophobic interactions with Trp855 (20.7%) and Thr861 (18.1%), complemented by strong hydrogen bonds with Trp855 (20.7%), Asp860 (38.4%), and Arg964 (33.4%). In contrast, mol11’s interactions are more focused, primarily involving hydrophobic contacts with Lys782 (13.2%) and Tyr1158 (11.8%), alongside a specific hydrogen bond with Lys782 (13.1%). mol12 utilizes hydrophobic interactions with Trp855 (17.5%) and Ile1159 (8.1%), and a significant hydrogen bond with Val856 (21.5%). mol13 demonstrates substantial hydrophobic engagement with Leu801 (15.2%) and Ala864 (17.8%), supported by an extensive network of hydrogen bonds, including strong associations with Lys782 (19.0%), Gln783 (13.4%), Lys803 (12.7%), and Val 856 (10.6%).

Among the evaluated compounds, mol14 exhibits the most compelling interaction profile with mTOR. Its hydrophobic interactions are exceptionally pronounced and broad, with Leu801 demonstrating an outstanding 35.8% occupancy, further supported by significant contributions from Ile853 (23.0%), Trp855 (18.0%), Thr861 (15.1%), and Ile972 (16.8%). This extensive hydrophobic binding suggests a superior fit within the mTOR hydrophobic pockets. Moreover, mol14 forms an exceptionally strong and diverse network of hydrogen bonds, notably with Asp811 and Asp973, both showing a remarkable 40.0% engagement, along with Lys803 (27.4%), Trp855 (18.0%), and Arg964 (16.0%). The dual role of Trp855 in both hydrophobic and hydrogen bonding interactions highlights its importance as a key residue for mol14 binding. The unprecedented strength and breadth of these combined interactions indicate a high binding stability and stability for mol14 with mTOR. This comprehensive interaction profile positions mol14 as the most promising candidate for further development as a potent mTOR inhibitor, surpassing the observed interactions of rapamycin, everolimus, and the other novel ligands investigated.

In summary, the interaction heatmaps reveal that mol10 and mol14 form the most persistent and structurally coherent contacts with mTOR, mirroring the stability observed in their dynamic and energetic profiles (Figure 12). These ligands display strong multivalent anchoring and a high degree of residue specificity, supporting their potential as robust mTOR modulators. On the other hand, mol11 shows limited contact permanence, underscoring a weaker binding mode. These findings complement the RMSD, RMSF, MM/PBSA, and PCA analyses, providing a comprehensive temporal perspective on ligand–receptor interaction stability.

### 2.5. Results of ADME Filters

SwissADME profiling (Appendix A) supports the suitability of the lignin-derived series as mTOR inhibitor candidates. Key physicochemical descriptors fall in the same regime as rapamycin and everolimus—e.g., high TPSA (≈199–249 Å^2^ for lignin oligomers vs. 195–205 Å^2^ for the controls) and macrocyclic-scale molecular weights (~691–725 g/mol)—while the consensus log Po/w spans a moderate range (−0.40 to 2.84), favoring aqueous compatibility and potentially reducing nonspecific partitioning relative to the more lipophilic references (4.19–4.62). PAINS screening is clean for two candidates (mol10, mol11), and structural alerts are minimal for mol10 (no Brenk flags) and limited for mol11 (single hydroquinone); notably, the clinical comparators also carry alerts, indicating that the alert burden of the best lignin analogs is not greater than that of approved chemotypes. Collectively, these features depict a developability profile consistent with effective mTOR inhibitors and highlight mol10, and, more broadly, the lignin series, as promising starting points for further optimization.

### 2.6. Multi-Observable Prioritization of Lignin-Derived Ligands

We prioritized ligands by integrating all analyses performed up to this point: MM/PBSA ΔG, PCA compactness, time-resolved interaction occupancy (standardized notation), and RMSD/RMSF. A concise summary is provided in Table 2.

Specifically, we integrated orthogonal observables: (i) MM/PBSA binding free energies computed on 8000 frames from the last 800 ns of each 1 µs trajectory, (ii) PCA of ligand heavy atoms to quantify conformational sampling, (iii) time-resolved interaction persistence (hydrogen bonds and hydrophobic contacts), and (iv) RMSD/RMSF stability mapping. On this multi-criteria basis, mol14 consistently ranks top (ΔG = −37.76 ± 3.41 kcal/mol; most compact PCA projection; broad, persistent network including Leu801 (35.8%), Ile853 (23.0%), Trp855 (18.0%), Thr861 (15.1%), Ile972 (16.8%), and strong H-bonds to Asp811/Asp973 (~40%)), indicating a highly stable and specific binding mode. mol10 also performs robustly, combining a very low, flat ligand RMSD (~0.20–0.25 nm) with a compact PCA footprint and persistent contacts/H-bonds (Trp855, Asp860, Arg964). By contrast, mol11 remains dynamically unstable (dispersed PCA; no RMSD plateau), and mol13, while energetically favorable (ΔG ≈ −31.0 kcal/mol), explores multiple basins and undergoes late rearrangements; mol12 exhibits a two-state behavior with a stable second plateau. Overall, the integrated metrics identify mol14 as the lead candidate and mol10 as a strong runner-up, whereas mol13’s favorable ΔG is tempered by conformational heterogeneity, mol12 stabilizes only after a late state transition, and mol11 remains the least promising due to persistent instability. The present prioritization is based solely on MD-derived observables and MM/PBSA estimates; no experimental validation is available yet. These results should therefore be regarded as hypothesis-generating to guide forthcoming binding and cellular assays.

## 3. Materials and Methods

The bioinformatics workflow employed in this study and described in Figure 13 follows a hierarchical approach to investigate the interactions between mTOR and specific ligands. It integrates molecular dynamics simulations, binding free energy calculations, principal component analysis (PCA), and detection of specific ligand–residue interactions. All primary simulations described in this study were run on a high-performance computing (HPC) infrastructure, enabling the generation of multiple complex trajectories and efficient handling of the large data volumes produced.

### 3.1. Protein Structure Preparation

This study began with the structural preparation of mTOR, based on its available crystallographic models (PDB IDs: 4JSX) [5]. The crystal structures of mTOR were of relatively low resolution (R-factor > 3 Å), resulting in poorly defined regions within the protein scaffold, particularly in flexible loop domains. Accurate reconstruction of the missing segments was essential for reliable molecular dynamics simulations, and homology modeling is a widely accepted approach to address such issues in structural bioinformatics and drug design [41]. Homology modeling was performed using RoseTTAFold via the Robetta server [42,43], based on the UniProt sequences of mTOR. This approach preserved the experimentally resolved regions from the X-ray structure while predicting plausible conformations for the missing loops, resulting in a complete and structurally coherent starting model. On the protein, we prioritized a single, functionally critical site; exploration of additional allosteric sites is therefore deferred to future work.

To validate the binding site, we used FPocketWeb (v1.0.1) [44], obtaining a binding-site volume of 378.0 Å^3^ and a solvent-accessible surface area of 110.1 Å2.

### 3.2. Preprocessing of Molecular Dynamics Simulations

The initial three-dimensional (3D) molecular structure of the ligands was generated from its Simplified Molecular Input Line Entry System (SMILES) string using Open Babel [45]. Subsequently, the geometry of the generated structure was optimized through energy minimization employing the General AMBER Force Field (GAFF) within Avogadro software (version 1.100) [46].

Exploration of additional allosteric sites is therefore deferred to future work. The resulting optimized structure, in Protein Data Bank (PDB) format, was then converted to the PDBQT format using the Python Molecule Viewer (PMV, version 1.5.7) suite, which includes AutoDockTools [47]. This PDBQT format incorporates partial charges and atom types compatible with AutoDock programs.

To ensure conformational stability prior to docking, the protein structure was subjected to energy minimization followed by short relaxation molecular dynamics simulation using GROMACS 2023.3 [48]. Molecular docking was then performed with AutoDock Vina 1.1.2 [49,50]. The protein was docked against a panel of seven ligands: rapamycin, mol10, mol11, mol12, mol13, mol14, and everolimus.

To increase robustness and mitigate single-run scoring variability, we performed 25 independent docking runs per protein–ligand complex, consistent with existing docking recommendations [51].

This step allowed the identification of the most favorable binding positions to be used as starting structures for molecular dynamics simulations.

### 3.3. Molecular Dynamics Simulations

To further assess the stability of the docked complexes, the top-ranked ligand–protein poses from the docking step of each complex—7 in total—were each subjected to a 1 μs (1000 ns) molecular dynamics (MD) simulation. All simulations were performed using GROMACS 2023.3 with the AMBER99SB-ILDN force field [52,53]. Topologies for each compound were generated using Acpype v2023.10.27 [54], which assigns AMBER atom types. The 7 ligand–receptor complexes obtained from the docking study were used as starting structures for the MD simulations. Each complex was placed at the center of the simulation box, solvated with TIP3P water molecules [55], and neutralized by adding appropriate counterions. For completeness, the composition of each solvated and neutralized simulation box used in the MD setup (number of water molecules, net charge before neutralization, and counterion species/counts) is reported in Appendix A. Energy minimization was performed using the steepest descent algorithm, allowing up to 50,000 steps and terminating when the maximum force dropped below 1000 kJ·mol^−1^·nm^−1^. The system was first equilibrated for 1 ns in the NVT ensemble at 300 K using the V-rescale thermostat, followed by 2 ns of NPT equilibration using the C-rescale barostat set to 1 bar. Long-range electrostatics were computed using the GPU-accelerated particle-mesh Ewald (PME) method for each protein–ligand complex. Each equilibrated system was then subjected to a 1 μs production run, integrated with a 2 fs timestep, with coordinates saved every 10 ps. The resulting trajectories were analyzed to characterize the time-dependent behavior of protein–ligand interactions. Two primary analyses—root mean square deviation (RMSD) and root mean square fluctuation (RMSF)—were performed to assess the overall stability and residue-level flexibility of each complex.

### 3.4. Binding Free Energy Estimation and Principal Component Analysis

The MM/PBSA method [56,57,58,59] was used to estimate the binding free energies of the 7 complexes analyzed in this study. Calculations were performed on 8000 frames extracted from each 1 μs molecular dynamics trajectory, starting at 0.200 μs, to capture the most representative conformations of each protein–ligand complex. Principal component analysis (PCA) was employed to reduce the dimensionality of the simulation data. This widely used technique projects high-dimensional datasets onto a smaller set of orthogonal axes (principal components) that capture the majority of the variance. The gmx covar module from the GROMACS package was utilized to compute and diagonalize the covariance matrix based on the coordinates of the ligand’s heavy atoms (carbons and oxygens) based on the whole MD trajectory. The resulting eigenvectors and eigenvalues were then analyzed using the gmx anaeig module to characterize the dominant collective motions throughout the simulation.

### 3.5. Ligand–Receptor Interaction Analysis

Ligand–receptor interactions were identified using the MD-ligand–receptor pipeline [60], focusing on those maintained throughout the majority of the simulation time (last 800 ns). Interactions were classified by bond type and by their persistence over the course of the trajectory. This analysis enabled a comparative assessment of interaction stability across the different mTOR–ligand complexes.

### 3.6. Drug-Likeness and ADME Filters

Drug-likeness and early ADME properties were computed with the SwissADME web server (www.swissadme.ch, accessed on 18 August 2025) [61] for mol10–mol14 and for the clinical references rapamycin and everolimus. SMILES strings (Table 1) were submitted under default settings. We recorded rule compliance and violations for the Lipinski, Veber, Ghose, Egan, and Muegge filters; screened for PAINS and Brenk structural alerts; and extracted MW, TPSA, and the consensus log Po/w. Results were exported and are tabulated in Appendix A.

### 3.7. Computational Resources and Infrastructure

All simulations and downstream analyses reported in this work were executed on the EuroHPC pre-exascale Tier-0 supercomputer LEONARDO (https://www.hpc.cineca.it/systems/hardware/leonardo, accessed on 8 July 2025), ranked 7th in the 65th edition of the TOP500 list. LEONARDO comprises two main partitions: a Booster Module based on BullSequana XH2135 nodes (each equipped with one Intel CPU and four NVIDIA Tensor Core GPUs) and a Data-centric Module based on BullSequana X2140 three-node CPU blades (each node featuring two Intel Sapphire Rapids CPUs, 56 cores each). The system employs NVIDIA Mellanox HDR 200 Gb/s InfiniBand interconnect with in-network computing acceleration, delivering low latency and high throughput to support large-scale MD workloads and post-processing [37].

### 3.8. Limitations of the Computational Approach

It is important to acknowledge a key limitation inherent in the study’s design. The analysis and subsequent conclusions for each of the seven ligand–mTOR complexes are derived from a single, continuous 1 µs molecular dynamics simulation. While this long timescale offers significant insight into the stability and interaction dynamics of each system, a single trajectory may not fully sample the entire conformational space available to the complex. The possibility exists that a trajectory could explore a distinct, long-lived metastable state that is not fully representative of the thermodynamic ensemble.

Consequently, while the convergence observed across multiple metrics within this simulation provides a strong indication of stability for the observed binding poses, the quantitative findings, particularly the MM/PBSA free energy rankings, should be interpreted with this consideration in mind. The gold standard for ensuring robust conformational sampling involves running multiple, independent replica simulations for each system. Therefore, the results presented here should be viewed as a robust basis for generating hypotheses, providing a detailed characterization of one plausible and stable binding scenario for each ligand, which will require further validation through subsequent experimental assays or more extensive computational studies employing enhanced sampling or replication-based approaches.

### 3.9. Data Availability

Relevant files and plots generated as part of this analysis are made available online on a Figshare data repository (Table 3).

## 4. Conclusions

This work presents an integrated, MD-driven assessment of seven mTOR–ligand complexes (five lignin-derived oligomers plus rapamycin and everolimus) on a 1µs timescale. By jointly evaluating ligand/protein RMSD/RMSF, PCA of ligand heavy-atom coordinates, time-resolved interaction persistence (hydrogen bonds and hydrophobics), and MM/PBSA binding free energies, we derived a consistent ranking of candidates. Across observables, mol14 emerges as the lead ligand and mol10 as a strong runner-up; mol13 exhibits favorable ΔG but greater conformational heterogeneity, mol12 stabilizes after a late transition to a second state, and mol11 remains the least stable. The residue-level interaction patterns that underpin these trends are reported with standardized occupancies in Appendix A, the ADME profiles supporting developability are summarized in Appendix A, and the cross-metric prioritization is consolidated in Appendix A (system composition in Appendix A).

From a translational perspective, the results support lignin-derived oligomers as a sustainable scaffold for mTOR inhibition: the best performers combine compact conformational sampling, persistent multivalent contacts at the catalytic cleft, and ADME properties comparable to clinical comparators. These atomistic insights highlight specific hotspot residues and contact motifs that can guide structure–activity optimization and prioritization for synthesis/testing. This study is entirely computational and is therefore limited by the absence of experimental validation. Future work will address these points through biophysical binding and cellular assays to validate the ranking, apo simulations under matched protocols, and rigorous free energy calculations, alongside exploration of additional pockets for top candidates. Together, these steps will refine the computational hypotheses into experimentally grounded leads.

## Figures and Tables

**Figure 1 ijms-26-08728-f001:**
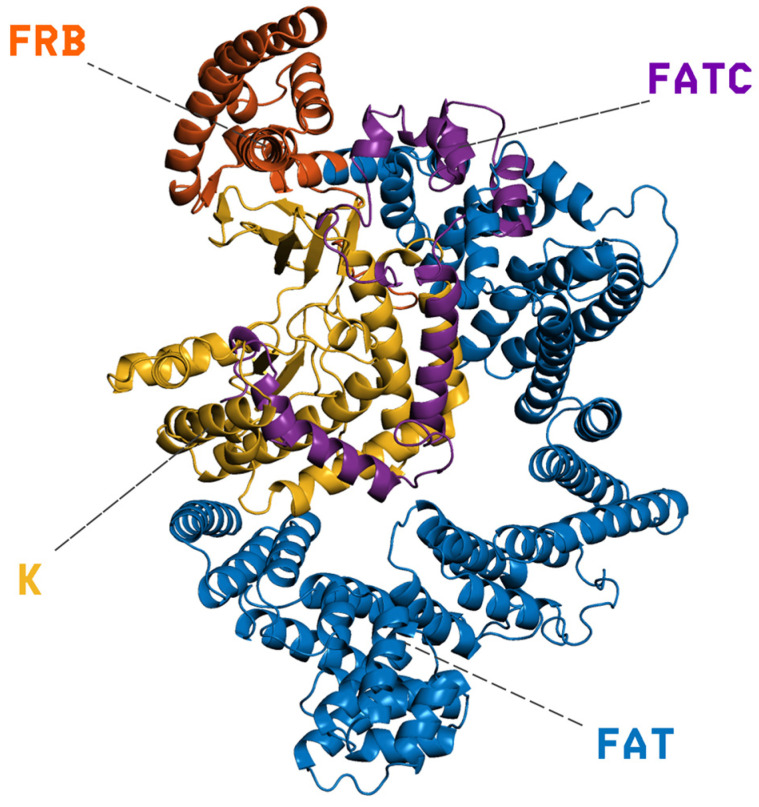
Cartoon representation of mTOR’s domain architecture: the extended N-terminal FAT α-helical scaffold (residues 1–629, blue) supports the FRB regulatory segment (residues 630–763, orange), followed by the catalytic kinase (K) domain of the PIKK family (residues 764–1045, yellow) and the short C-terminal FATC motif (residues 1046–1164, purple). Dashed leaders mark the labeled domains; the contiguous arrangement of FAT–FRB–K–FATC shown here illustrates how regulatory inputs are structurally transmitted to the active site.

**Figure 2 ijms-26-08728-f002:**
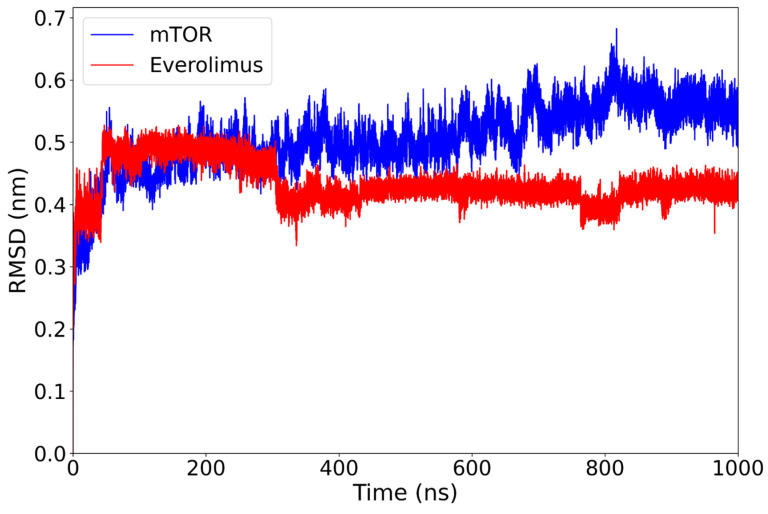
RMSD plot of the mTOR protein (blue) and everolimus ligand (red) over a 1000 ns molecular dynamics simulation. The protein structure shows a gradual increase from around 0.50 nm (400–600 ns) to approximately 0.55 nm (800–1000 ns) after the initial equilibration phase, while the ligand maintains a lower and relatively stable RMSD around 0.43 nm, indicating persistent binding stability within the mTOR active site.

**Figure 3 ijms-26-08728-f003:**
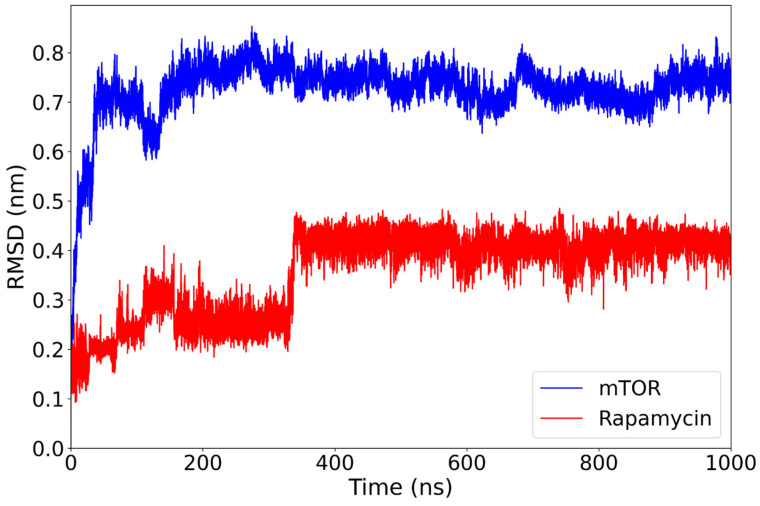
RMSD of the mTOR backbone (blue) and the rapamycin ligand (red) over the course of a 1000-nanosecond molecular dynamics simulation. The mTOR protein reaches structural stability after approximately 150 ns, maintaining RMSD values around 0.75–0.85 nm for the remainder of the trajectory. The rapamycin ligand exhibits a more gradual stabilization, with a distinct conformational transition occurring around 350 ns, after which its RMSD plateaus near 0.40 nm.

**Figure 4 ijms-26-08728-f004:**
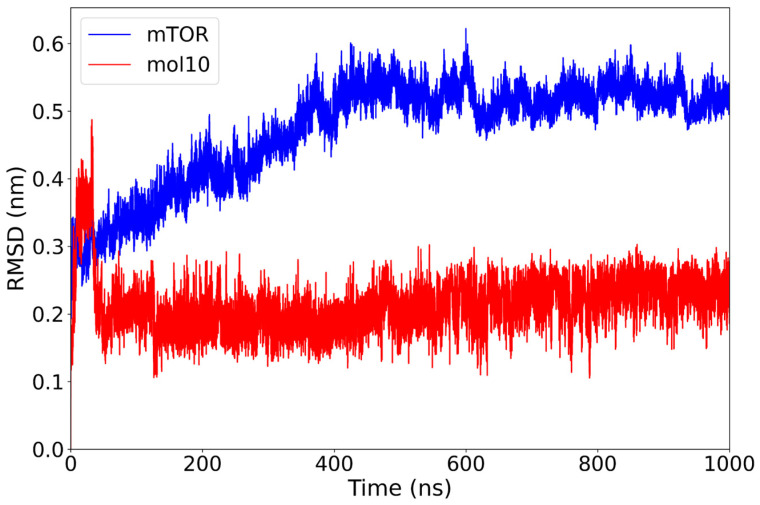
RMSD plot of the mTOR protein (blue) and mol10 ligand (red) over a 1000 ns molecular dynamics simulation. The ligand exhibits a consistently low and stable RMSD around 0.2 nm, suggesting a strong and persistent interaction with the binding site. In contrast, the mTOR protein undergoes a gradual increase in RMSD, stabilizing around 0.5 nm, indicative of conformational flexibility upon ligand binding.

**Figure 5 ijms-26-08728-f005:**
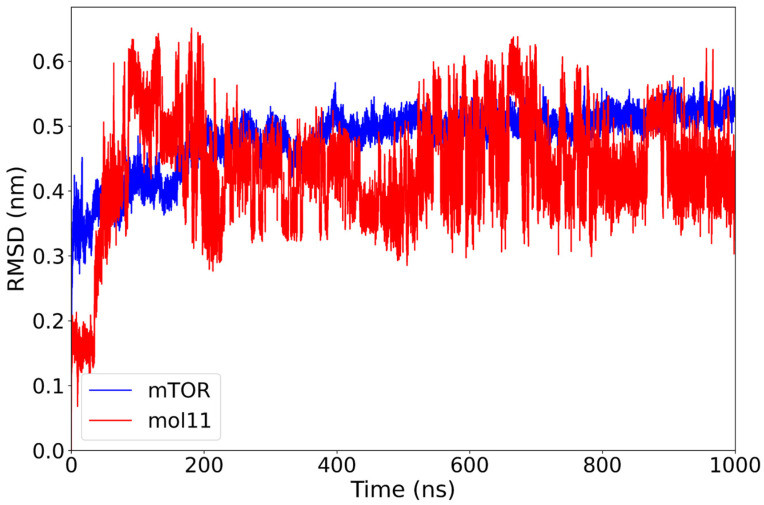
RMSD plot of the mTOR protein (blue) and mol11 ligand (red) over a 1000 ns molecular dynamics simulation. While the mTOR protein reaches stability after approximately 200 ns with an RMSD around 0.48 nm, the mol11 ligand exhibits high variability throughout the simulation, indicating an unstable binding mode and possible repositioning within the binding pocket.

**Figure 6 ijms-26-08728-f006:**
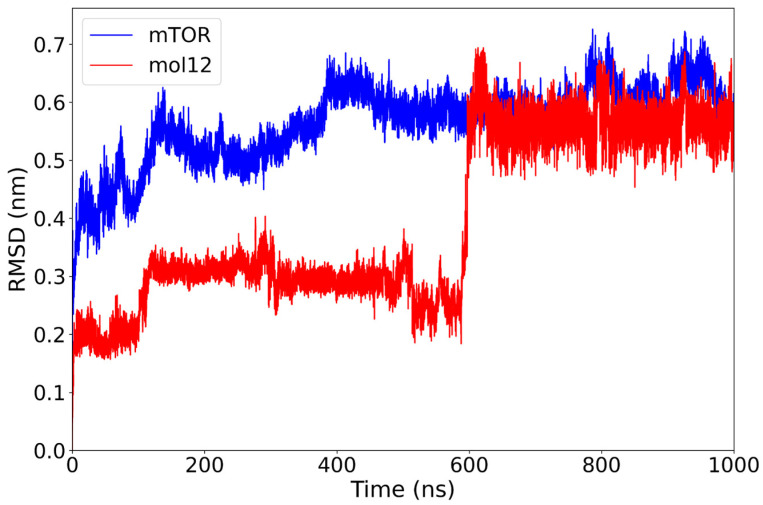
RMSD plot of the mTOR protein (blue) and mol12 ligand (red) over a 1000 ns molecular dynamics simulation. The mTOR protein exhibits stable behavior after approximately 130 ns with RMSD values around 0.6 nm. The mol12 ligand displays notable variability, especially around 600 ns, suggesting a potential conformational rearrangement or repositioning within the binding pocket.

**Figure 7 ijms-26-08728-f007:**
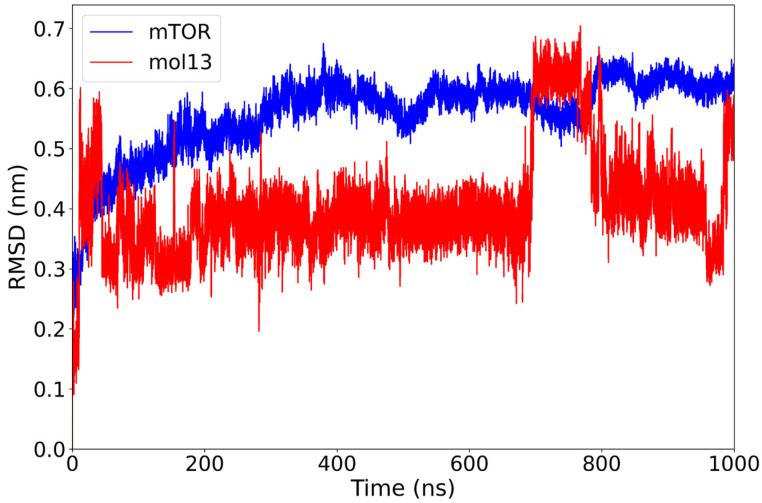
RMSD plot of the mTOR protein (blue) and mol13 ligand (red) over a 1000 ns molecular dynamics simulation. The mTOR protein shows consistent structural stability after the initial equilibration phase. The mol13 ligand exhibits marked variability throughout the final part of the simulation, with multiple rapid changes in RMSD values, indicating significant conformational rearrangements.

**Figure 8 ijms-26-08728-f008:**
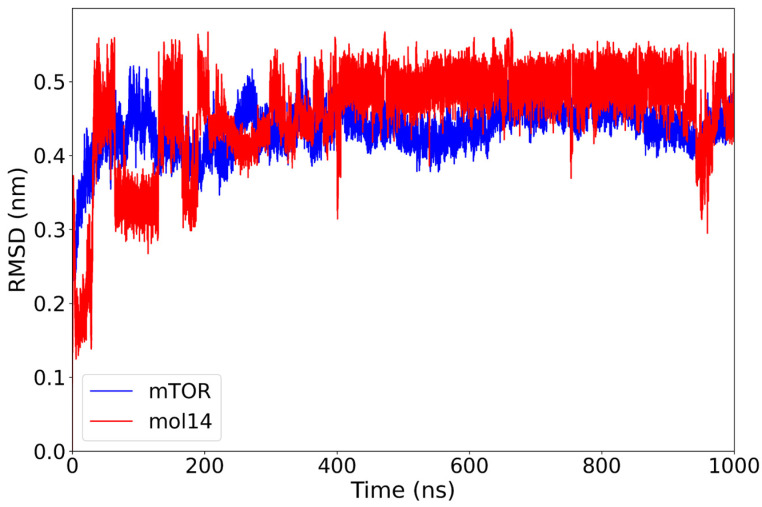
RMSD plot of the mTOR protein (blue) and mol14 ligand (red) over a 1000 ns molecular dynamics simulation. The mTOR protein rapidly equilibrates and maintains a stable RMSD around 0.45 nm after the initial 50 ns. The mol14 ligand exhibits initial mobility with moderate variability for the first 400 ns, after which it reaches a plateau at approximately 0.48 nm, suggesting a relatively stable binding mode characterized by some internal flexibility or minor rearrangements.

**Figure 9 ijms-26-08728-f009:**
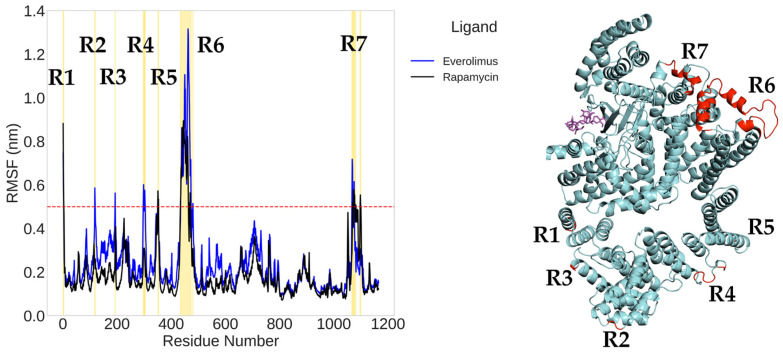
Root mean square fluctuation (RMSF) analysis of mTOR in complex with everolimus (blue) and rapamycin (black) during molecular dynamics simulations (**left**), and spatial mapping of the most mobile regions onto the 3D protein structure (**right**). The red dashed line at 0.5 nm represents the threshold above which residues are considered highly mobile. Seven distinct regions (R1–R7) exceeding this threshold are highlighted in yellow on the plot and mapped in red on the protein structure. The positions of everolimus and rapamycin are shown as magenta stick models, respectively, providing spatial context with respect to the flexible segments of the protein.

**Figure 10 ijms-26-08728-f010:**
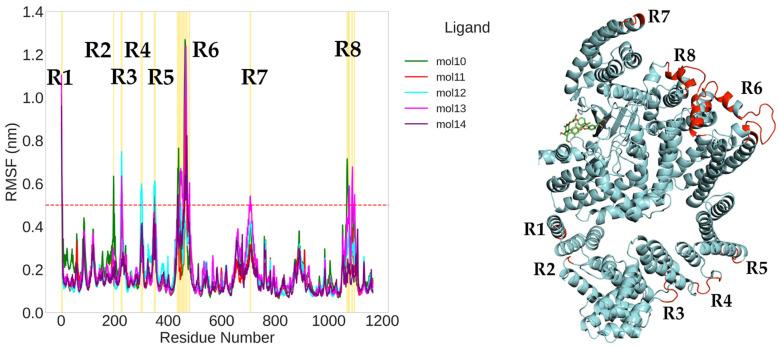
Root mean square fluctuation (RMSF) analysis of mTOR in complex with the small molecules mol10 to mol14 (**left**), and corresponding mapping of the most mobile protein segments onto the 3D structure (**right**). Colored lines represent the RMSF profiles for each complex, with a red dashed line indicating the 0.5 nm threshold for enhanced flexibility. Eight regions (R1–R8) exceeding this threshold are shaded in yellow on the plot and shown in red on the protein structure. The bound ligands are visualized as stick models in green, illustrating their spatial relationship to the fluctuating regions.

**Figure 11 ijms-26-08728-f011:**
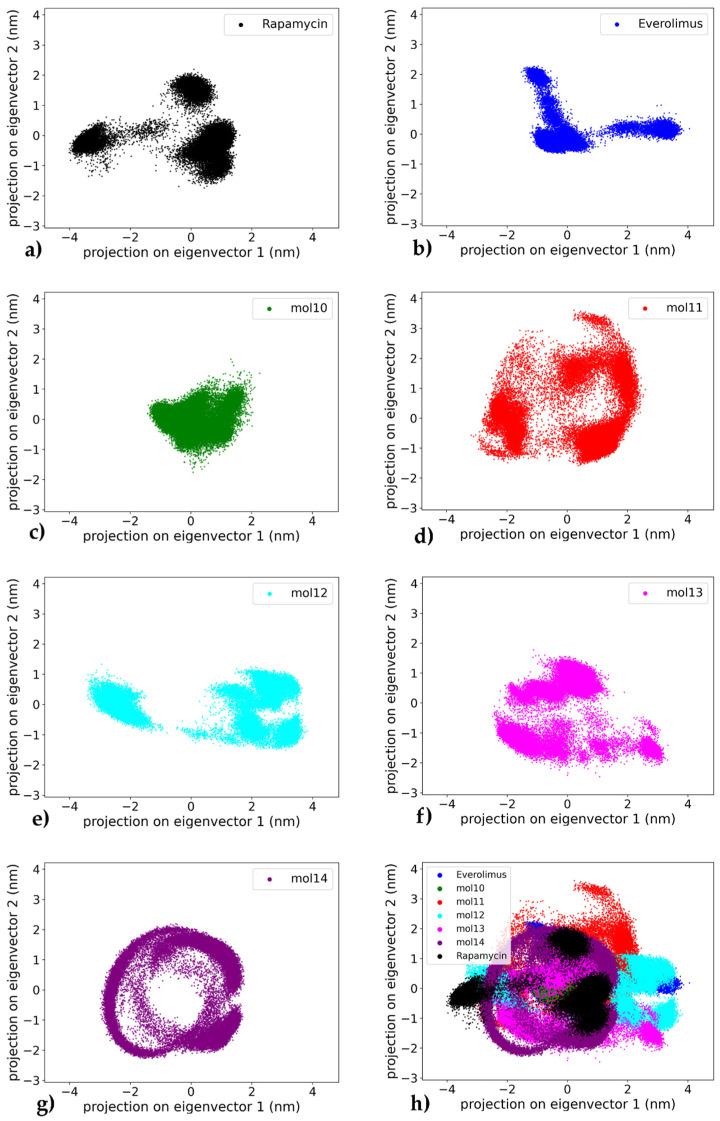
Comparative principal component analysis (PCA) of ligand conformational dynamics in complex with mTOR. Each panel shows the projection of ligand heavy-atom trajectories onto PC1–PC2 from 1000 ns MD; PCs were computed separately for each complex and the apo protein is not included: (**a**) Rapamycin—compact core with limited dispersion; (**b**) Everolimus—compact sampling with three small clusters; (**c**) mol10—narrow/compact sampling; (**d**) mol11—widest dispersion (high variability); (**e**) mol12—two moderately compact clusters; (**f**) mol13—fragmented, widespread sampling; (**g**) mol14—most compact, ring-like envelope; (**h**) Overlay of panels (**a**–**g**) to qualitatively compare compactness/dispersion across ligands (cross-color overlap reflects similar ligand conformational envelopes, not shared protein motions).

**Figure 12 ijms-26-08728-f012:**
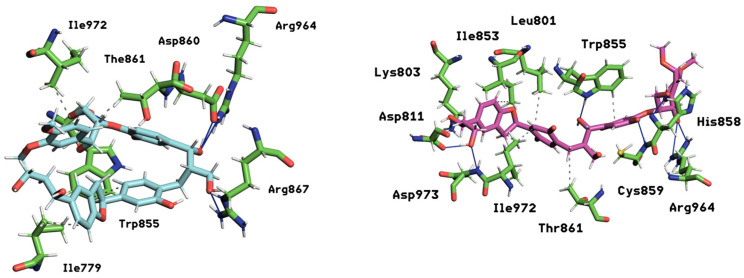
Key interactions between two ligands, mol10 (left ligand rendered in light blue) and mol14 (right ligand rendered in magenta), within the mTOR protein binding site. The surrounding protein residues involved in these interactions are shown in green sticks. Principal interactions are depicted, with hydrophobic interactions represented by dashed gray lines and hydrogen bonds by continuous blue lines, illustrating the nature of the binding pocket.

**Figure 13 ijms-26-08728-f013:**
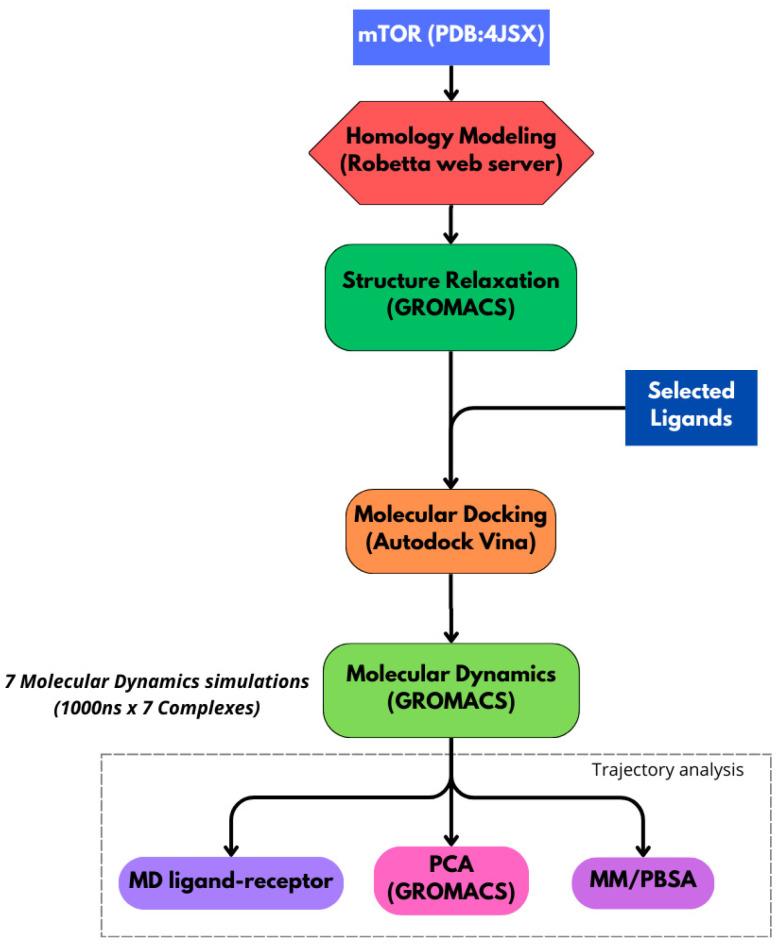
Schematic of the study workflow. We used the structure of mTOR (PDB IDs: 4JSX), with Robetta-based homology modeling. The structure was relaxed in GROMACS. Docking was performed with lignin-derived ligands and natural inhibitors rapamycin and everolimus. Complexes underwent 1000 ns GROMACS MD simulations. Analyses included ligand–receptor interactions, PCA, and MM/PBSA.

**Table 1 ijms-26-08728-t001:** Ligands of interest: 2D chemical structure and SMILES format.

Ligand Name	Ligand Structure	SMILES String
mol10	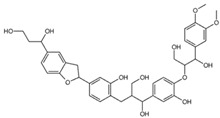	OCC(C(O)C1=CC=C(OC(CO)C(O)C2= CC=C(OC)C(OC)=C2)C(O)=C1)CC(C(O)= C3)=CC=C3C4CC5=CC(C(O)CCO)=CC=C5O4
mol11	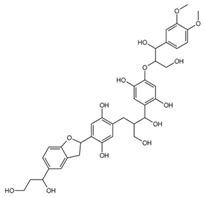	OCC(C(O)C1=C(O)C=C(OC(CO)C(O)C2= CC=C(OC)C(OC)=C2)C(O)=C1)CC(C(O)= C3)=CC(O)=C3C4CC5=CC(C(O) CCO)=CC=C5O4
mol12	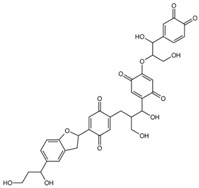	OCC(C(O)C(C(C=C1OC(CO)C(O)C(C= CC2=O)=CC2=O)=O)=CC1=O)CC3=CC(C (C4CC5=CC(C(O)CCO)=CC=C5O4)= CC3=O)=O
mol13	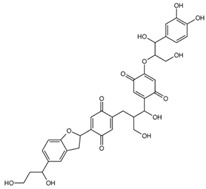	OCC(C(O)C(C(C=C1OC(CO)C(O)C2= CC(O)=C(O)C=C2)=O)=CC1=O)CC3=CC(C (C4CC5=CC(C(O)CCO)=CC=C5O4)= CC3=O)=O
mol14	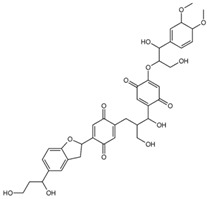	OCC(C(O)C(C(C=C1OC(CO)C(O)C2=CC (OC)C(OC)C=C2)=O)=CC1=O)CC3= CC(C(C4CC5=CC(C(O)CCO)=CC=C5O4)= CC3=O)=O
Rapamycin	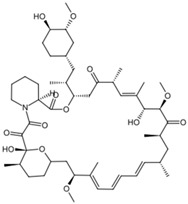	C[C@@H]1CCC2C[C@@H](/C(=C/C= C/C=C/[C@H](C[C@H](C(=O)[C@@H] ([C@@H](/C(=C/[C@H](C(=O)C[C@H](OC(=O) [C@@H]3CCCCN3C(=O)C(=O)[C@@]1(O2)O) [C@H](C)C[C@@H]4CC[C@H]([C@@H](C4) OC)O)C)/C)O)OC)C)C)/C)OC
Everolimus	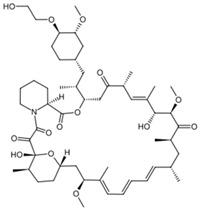	C[C@@H]1CC[C@H]2C[C@@H](/C(=C/C= C/C=C/[C@H](C[C@H](C(=O)[C@@H]([C@@H] (/C(=C/[C@H](C(=O)C[C@H](OC(=O)[C@@H] 3CCCCN3C(=O)C(=O)[C@@]1(O2)O)[C@H](C) C[C@@H]4CC[C@H]([C@@H](C4)OC)OCCO)C)/ C)O)OC)C)C)/C)OC

**Table 2 ijms-26-08728-t002:** Multi-observable prioritization of ligands. Values summarized from RMSD/PCA, MM/PBSA, and interaction-persistence analyses. Free energy and interaction analyses computed on 8000 frames from the last 800 ns of each 1 µs trajectory. ΔG values are reported as mean ± SD from MM/PBSA. Interaction persistence is reported as occupancy (%).

Ligand	MM/PBSA ΔG (kcal/mol)	Ligand RMSD (1 µs)	PCA (Ligand Heavy Atoms)	Most Persistent Interactions (Occupancy, %)
Rapamycin	−26.39 ± 6.14	Stable plateau ~0.40 nm after ~350 ns	Dense core, limited spread	Hydrophobics: ILE801 (8.1%), THR861 (6.9%), ILE972 (14.0%); H-bonds: THR780 (27.1%), GLN777 (28.0%).
Everolimus	−31.51 ± 9.61	Stable (~0.43 nm); protein ~0.47 nm	Compact clustering	Hydrophobics: LEU877 (15.4%), TYR1158 (21.8%); H-bond: LYS986 (26.1%).
mol10	−26.37 ± 4.84	Very low and flat (~0.20–0.25 nm)	Narrow/compact	TRP855 (20.7%) hydroph./H-bond; H-bonds: ASP860 (38.4%), ARG964 (33.4%).
mol11	−25.93 ± 6.44	High variability, no plateau (>0.6 nm)	Widest dispersion	Hydrophobics: LYS782 (13.2%), TYR1158 (11.8%); H-bond: LYS782 (13.1%).
mol12	−25.12 ± 7.54	One transition ~600 ns → stable plateau ~0.55 nm	Two moderately compact clusters	Hydrophobics: TRP855 (17.5%), ILE1159A (8.1%); H-bond: VAL856 (21.5%).
mol13	−31.02 ± 7.51	Multiple late rearrangements (0.35→0.6→0.4–0.5 nm)	Wide/fragmented cluster	Hydrophobics: LEU801A (15.2%), ALA864A (17.8%); H-bonds: LYS782A (19.0%), GLN783A (13.4%), LYS803A (12.7%), VAL856A (10.6%).
mol14	−37.76 ± 3.41	Reaches plateau ~0.48 nm by ~400 ns	Most compact; ring-like, restricted drift	Hydrophobics: LEU801 (35.8%), ILE853 (23.0%), TRP855 (18.0%), THR861 (15.1%), ILE972 (16.8%); H-bonds: ASP811 (40.0%), ASP973 (40.0%), LYS803 (27.4%), TRP855 (18.0%), ARG964 (16.0%).

**Table 3 ijms-26-08728-t003:** Listing of data files made available on the Figshare portal.

Item	Data Type	Accession
Ligands	PDB files	https://doi.org/10.6084/m9.figshare.29598395, accessed on 18 August 2025
Protein/MD starting coordinates	mTOR_protein.pdb file	https://doi.org/10.6084/m9.figshare.29646449, accessed on 18 August 2025
Ligand–receptor analysis	CSV and PNG files	https://doi.org/10.6084/m9.figshare.29598389, accessed on 18 August 2025
PCA plots	PNG files	https://doi.org/10.6084/m9.figshare.29598419, accessed on 18 August 2025
RMSD and RMSF plots	PNG files	https://doi.org/10.6084/m9.figshare.29598422, accessed on 18 August 2025
MM/PBSA free binding energy	CSV file	https://doi.org/10.6084/m9.figshare.29603588, accessed on 18 August 2025

## Data Availability

All produced data are available on figshare (see Table 3).

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
