# Peer review of "Lignin-Derived Oligomers as Promising mTOR Inhibitors: Insights from Dynamics Simulations"

_ijms, 2025, doi:10.3390/ijms26178728_

Round 1

Reviewer 1 Report

Comments and Suggestions for Authors

The manuscript entitled " Lignin-Derived Oligomers as Promising mTOR Inhibitors: Insights from Dynamics Simulations " by Sofia Gabellone, et. al., explained about the manuscript has been written regular. There are some revisions for better understanding as below: 

  1. In Fig. 2, the authors state that this transient change does not compromise structural stability. However, a supporting literature reference should be provided at this point.
  2. Why does the protein group in Fig. 2 exhibit RMSD values that differ markedly from those shown in Figs. 1, 3–7, even though all plots ostensibly track the same system?
  3. The abbreviation “RMSD” is used inconsistently throughout the manuscript. Please standardize the term to “RMSD” upon first mention and use only the abbreviation thereafter.
  4. The authors should format all paragraphs in the Results and Discussion section to match the style used in the Introduction.
  5. In Fig. 4, why does the ligand RMSD fluctuate so dramatically? The authors should provide a clear explanation.
  6. The degree of interpretation should be noted in the PCA analysis.
  7. Why is a threshold of 0.5 nm used to identify regions with enhanced dynamic behavior in root mean square fluctuation (RMSF) analysis? Is there literature support for this?
  8. Normally, a combination of computational and experimental approaches provides more robust validation of results. How does the manuscripts address computational false positives to ensure the reliability of findings?
  9. Please pay attention to the notation for amino acids in Figure 11, such as Arg964.
  10. Standardize the annotation format, Figure 1. or Figure 1:.

Reviewer 2 Report

Comments and Suggestions for Authors

Summary

This manuscript explores the potential of five lignin-derived oligomers (mol10–mol14) as novel mTOR inhibitors, comparing them to Rapamycin and Everolimus using a comprehensive computational approach. The authors perform 1000 ns all-atom MD simulations for each protein–ligand complex, followed by analyses including RMSD, RMSF, MM/PBSA binding energy, principal component analysis (PCA), and dynamic contact profiling. The work is contextualized within sustainable drug design and proposes these lignin fragments as dual-acting agents with delivery and inhibitory roles. The article is well-structured, with data made available on Figshare, enhancing reproducibility.

Positive aspects

 This is a scientifically solid and methodologically thorough manuscript that successfully integrates molecular dynamics with free energy estimations and conformational analyses. The study stands out in its systematic comparison between natural products and clinical inhibitors of mTOR. The use of a 1 µs MD simulation for each complex is commendable.

Minor Concerns:

  1. In section 3.2, the reference to "Open Babel[39]" should include a space before the bracket: "Open Babel [39]".
  2. In section 3.1, the citation "RoseTTAFold via the Robetta server [37][38]" should be revised for clarity.
  3. The manuscript inconsistently uses MM/PBSA and MM-PBSA. Please standardize terminology and clarify whether entropy was considered
  4. The binding site for docking is assumed based on prior studies but no pocket validation (e.g., via SiteMap, fpocket, or experimental comparison) is presented. This is critical since lignin fragments may favor alternative allosteric sites.
  5. Several plots, particularly RMSD/RMSF graphs and PCA projections (e.g., Figures 1-10), would benefit from enhanced visual clarity. The axis labels, tick marks, and legends are currently too small, which may hinder readability.
  6. The Methods section would benefit from additional detail to improve reproducibility, particularly regarding the Principal Component Analysis (PCA). For instance, it is not clearly stated which atom selection was used for PCA calculations (e.g., ligand heavy atoms, backbone atoms of the protein, or a combined subset).

Major Concerns:

  1. The study is based on single 1 µs MD trajectories per complex. Without replicates, conclusions regarding ligand stability and binding cannot account for potential entrapment in local minima or stochastic artifacts. Given the variability observed (e.g., mol13's multiple transitions), this is a significant limitation. At least two additional replicates for top candidates (e.g., mol14 and mol10) would strengthen statistical confidence.
  2. The docking step uses AutoDock Vina, but there is no mention of redocking to validate the scoring accuracy or pose prediction using the known crystallographic pose of Rapamycin. A redocking control would clarify if the docking box and scoring function correctly reproduce the native pose, an essential prerequisite for trusting subsequent MD simulations.
  3. PCA clustering and contact occupancy are overinterpreted as indicators of binding affinity or efficacy. These metrics indicate stability and sampling but should not be used alone to claim superiority in binding without complementary experimental data or free energy perturbation (FEP) methods
  4. While the study successfully identifies two lignin-derived compounds (mol10 and mol14) with favorable binding profiles to mTOR, potentially outperforming the clinical inhibitors Rapamycin and Everolimus, it lacks an initial assessment of their drug-likeness and pharmacokinetic/pharmacodynamic (PK/PD) profiles. Given the central therapeutic role of mTOR across multiple disease contexts, evaluating whether these compounds adhere to basic medicinal chemistry filters is crucial before advancing them as viable leads. The authors are encouraged to compute and report key drug-likeness and ADME-related rule violations, including Lipinski’s Rule of Five, Veber, Ghose, Egan, and Muegge filters, as well as perform PAINS (pan-assay interference compounds) and Brenk structural alert screening. These analyses would provide an early indication of oral bioavailability, promiscuity, and developability, allowing for a more balanced comparison with the clinically approved controls.
  5. The manuscript lacks a dedicated “Conclusions” section, which is essential in summarizing the key findings, their implications, and future directions. While the discussion provides detailed analysis and comparative insights, it does not culminate in a concise, high-level synthesis of the study’s outcomes. A well-crafted conclusions section would: reinforce the main computational findings, reflect on the potential translational impact, address limitations of the study.
  6. It is possible that experimental binding affinity data (e.g., IC₅₀ or Kd values) for Rapamycin and Everolimus against mTOR are available in the literature. If such data exist, the manuscript currently does not present any quantitative comparison between these experimental values and the ΔG estimates obtained from MM/PBSA calculations. Such a comparison would be critical to evaluate whether the computational protocol reproduces the correct order of magnitude and relative ranking for known inhibitors before extrapolating predictions to novel lignin-derived compounds. Even a brief discussion of this aspect, highlighting agreement, discrepancies, or lack of available experimental data, would significantly strengthen the credibility of the computational approach and its predictive power.
  7. The manuscript does not discuss potential drug developability limitations related to the physicochemical properties of the lignin-derived oligomers. Some of these compounds appear to be relatively large and highly polar, characteristics that could adversely affect their aqueous solubility, membrane permeability, and overall oral bioavailability. In addition, excessive molecular size may hinder effective access to the binding pocket, particularly if the FRB domain or adjacent regions impose steric constraints. Without addressing these considerations, the proposal to develop these oligomers as viable mTOR inhibitors may be overestimated. A brief discussion of these factors, ideally supported by basic in silico ADMET assessments (e.g., predicted logP, TPSA, molecular weight) or known trends for lignin-derived molecules, would provide a more balanced evaluation of their therapeutic potential.

Reviewer 3 Report

Comments and Suggestions for Authors

Major Comments

  1. Page 1: The sentence “A key component … such dysregulation” needs to be revised for clarity and improved readability.
  2. The introduction would benefit from a figure showing the mTOR kinase domain, with all key regulatory regions clearly labeled. Related structural discussion, currently scattered throughout the manuscript (e.g., on pages 9 and 10), should be integrated into the Introduction to provide a cohesive narrative.
  3. Only a single replica per ligand has been simulated. For enhanced statistical reliability, multiple independent replicas per ligand should ideally be performed. I strongly encourage authors to perform at-least 3 replicas for each system.
  4. The manuscript lacks details on the docking procedure. This information should be included either in the main text or as part of the Supporting Information.
  5. RMSD is a degenerate observable for ligand–protein binding, as it is associated with large entropy loss and does not account for conformational diversity or specific interactions. The selection of mol10, mol12, and mol14 as promising ligands based solely on RMSD requires justification. It is possible that the remaining ligands might require longer production times to fully stabilize within the binding pocket.
  6. The statement on pages 10–11—“The mammalian target of rapamycin pathway … developing novel mTOR inhibitors”—repeats a point already made in the final paragraph of the Introduction. This redundancy should be removed or reworded.
  7. Figures 8 and 9 need improved clarity. The font size of axis labels and legends should be increased for better readability, and the background grid can be removed for a cleaner visual presentation.
  8. The RMSF profiles of ligand-bound systems should be compared with those of the apo protein and known ligand-bound systems to assess how ligand binding influences protein flexibility.
  9. The application of MM/PBSA should be justified in light of concerns raised by Roux et al. (https://doi.org/10.1021/acs.jpcb.4c06614) regarding the method’s limitations, particularly its “end-point” approximation and use of implicit solvent models.
  10. Sections 2.2 and 2.3 currently share the same heading. This should be corrected for clarity and structure.
  11. The manuscript should include data and discussion for the apo mTOR system, including RMSD, RMSF, and principal component analysis (PCA), to provide a baseline for comparison with ligand-bound systems.
  12. The manuscript should specify how PCA was performed: which atoms were selected (e.g., Cα atoms, backbone atoms), and what frame range or trajectory segments were used in the analysis.
  13. The interpretation of PC1 vs PC2 scatter plots should be elaborated. For instance, if scatter plots from different systems (e.g., apo and ligand-bound) show overlapping clusters (e.g., in Fig. 10h), does this suggest that the protein samples similar conformational landscapes?
  14. On page 14, clarify what the percentage values in parentheses represent. Also, ensure that residues are consistently named using standard three-letter or one-letter amino acid codes followed by the residue number.

Minor Comments

  1. Page 7: The term “600 nm” should be corrected to “600 ns” as it refers to simulation time.

Reviewer 4 Report

Comments and Suggestions for Authors

The manuscript under consideration is based on a very significant number of calculations, so it is surprising that it does not even mention the computing resources with which they were performed.

Unfortunately, the significance of the conclusions made on the basis of these calculations is significantly inferior to their number. The main drawback of this work is the complete lack of comparison of the calculations with any experimental data.

If experimental data on the structure of the complexes are indeed absent, the authors should have considered several binding sites for each ligand, and not limited to one. It is known that the difference in energies for different variants may not be very significant. Also it should be borne in mind that after performing molecular dynamics, the ranking by binding energy may change. That is, the study would be much more useful if the authors considered not 7 different ligands, but 7 different docking sites for any of them. Accordingly, I suggest the authors choose any ligand, consider at least a couple more docking sites for it, and present the obtained results in the form of the first article. Then gradually analyze other ligands and write other articles.

In addition, it seemed that the authors performed different analytical procedures (RMSD calculation, MM/PBSA, etc.) in parallel and did not compare the results obtained. Otherwise, it is difficult to understand why 800 ns trajectory paths were taken for the MM/PBSA analysis, although Figures 1–7 (RMSD) show that most of the systems under consideration continue to change noticeably after 200 ns. If the authors wanted to use trajectories of the same length for equilibrium systems, then the last 200 ns should have been taken.

Further comments are more specific.

The manuscript contains many abbreviations, some of which are not deciphered at all (e.g., HEAT, FKBP12), and some (e.g., FAT, MM/PBSA) are deciphered many pages after the first mention, and the abbreviation MM/PBSA is written in several ways: MM/PBSA, MM-PBSA, MMPBSA, and in the list of abbreviations at the end of the article it is given with a typo (MM/GBSA). Now this list includes fairly well-known abbreviations (MD, RMSD), while specific biological abbreviations for this article are absent, although their meaning is almost impossible to guess. Accordingly, in my opinion, all biological abbreviations should be included in the list.

The authors write (page 3–4) that in the mTOR–Everolimus complex RMSD 'varies moderately', however, in my opinion, Fig. 1 clearly shows that the RMSD of the mTOR protein gradually increases with time: in the trajectory section from 400 to 600 ns it is approximately 0.5 nm, and in the section from 800 to 1000 ns it is already 0.55 ns, i.e. clearly greater than 0.47 nm, which the authors write about. I believe that the authors should make adjustments to the description of Figure 1.

The title of Section 2.3 repeats the title of Section 2.2, although the sections are devoted to different methods.

The molecular compositions of the systems described incompletely and not entirely clear (Section 3.3). Quote: 'Each complex was placed at the center of the simulation box, solvated with TIP3P water molecules [47], and neutralized by adding appropriate counterions.' It would be useful to indicate the number of water molecules in each system and to clarify which fragments required counterions (the ligands, judging by the structural formulas in Table 1, are neutral), which ones and how many.

The phrase 'Sustainable drug design', present in the keywords, does not reflect the content of this manuscript.

It is advisable to end the article with a section that briefly outlines the main results of the study.

Round 2

Reviewer 1 Report

Comments and Suggestions for Authors

The manuscript is acceptable for publication in its current form.

Author Response

Thank you

Reviewer 2 Report

Comments and Suggestions for Authors

In my view, the manuscript is now suitable for acceptance. The authors have addressed the reviewers’ comments with rigor and transparency, substantially improving the clarity and scientific value of the work. Although Major Comment 1 (absence of replicate MD simulations) remains unresolved, the authors provided a reasonable justification regarding computational resource constraints. I recommend that the manuscript be accepted provided the authors add a short dedicated paragraph or subsection immediately before the “Conclusions” to explicitly acknowledge this limitation. Such a statement will ensure that readers are aware of the potential bias introduced by relying on single long MD trajectories and will appropriately contextualize the robustness of the findings.

Author Response

We have added the requested part by inserting a new subsection 3.8 in Section 3, entitled “Limitations of the Computational Approach.”

Reviewer 3 Report

Comments and Suggestions for Authors

Authors have responded to all my queries positively. I do not have further comments, and manuscript can be accepted for publication in its current form.

Author Response

Thank you.

Reviewer 4 Report

Comments and Suggestions for Authors

The authors took into account those comments that could have been taken into account without additional calculations, and this, in my opinion, significantly improved the article. Although the remarks related to additional calculations, including the choice of other parts of molecular dynamic trajectories for further analysis, remained unaccounted for, I do not object to the publication of the material in the present form.

Author Response

Thank you.